# Synergistic Effects of Weight Loss and Catheter Ablation: Can microRNAs Serve as Predictive Biomarkers for the Prevention of Atrial Fibrillation Recurrence?

**DOI:** 10.3390/ijms25094689

**Published:** 2024-04-25

**Authors:** Carola Y. Förster, Stephan R. Künzel, Sergey Shityakov, Stavros Stavrakis

**Affiliations:** 1Department of Anaesthesiology, Intensive Care, Emergency and Pain Medicine, University of Würzburg, 97080 Würzburg, Germany; 2Institute for Transfusion Medicine, Faculty of Medicine Carl Gustav Carus, Technische Universität Dresden, 01307 Dresden, Germany; stephan.kuenzel@tu-dresden.de; 3Institute for Transfusion Medicine, German Red Cross Blood Donation Service North-East, 01307 Dresden, Germany; 4Laboratory of Chemoinformatics, Infochemistry Scientific Center, ITMO University, 197101 Saint-Petersburg, Russia; shityakoff@hotmail.com; 5Cardiovascular Section, Department of Medicine, University of Oklahoma Health Sciences Center, Oklahoma City, OK 73104, USA

**Keywords:** atrial fibrillation, catheter ablation, cytokine, fibrosis, hypertension, inflammaging, microRNA, obesity, recurrent atrial fibrillation, ROS, sex differences, weight loss

## Abstract

In atrial fibrillation (AF), multifactorial pathologic atrial alterations are manifested by structural and electrophysiological changes known as atrial remodeling. AF frequently develops in the context of underlying cardiac abnormalities. A critical mechanistic role played by atrial stretch is played by abnormal substrates in a number of conditions that predispose to AF, including obesity, heart failure, hypertension, and sleep apnea. The significant role of overweight and obesity in the development of AF is known; however, the differential effect of overweight, obesity, cardiovascular comorbidities, lifestyle, and other modifiable risk factors on the occurrence and recurrence of AF remains to be determined. Reverse remodeling of the atrial substrate and subsequent reduction in the AF burden by conversion into a typical sinus rhythm has been associated with weight loss through lifestyle changes or surgery. This makes it an essential pillar in the management of AF in obese patients. According to recently published research, microRNAs (miRs) may function as post-transcriptional regulators of genes involved in atrial remodeling, potentially contributing to the pathophysiology of AF. The focus of this review is on their modulation by both weight loss and catheter ablation interventions to counteract atrial remodeling in AF. Our analysis outlines the experimental and clinical evidence supporting the synergistic effects of weight loss and catheter ablation (CA) in reversing atrial electrical and structural remodeling in AF onset and in recurrent post-ablation AF by attenuating pro-thrombotic, pro-inflammatory, pro-fibrotic, arrhythmogenic, and male-sex-associated hypertrophic remodeling pathways. Furthermore, we discuss the promising role of miRs with prognostic potential as predictive biomarkers in guiding approaches to AF recurrence prevention.

## 1. Introduction

AF is the most important arrhythmia in clinical practice. AF frequently leads to stroke or heart failure, thus representing a major healthcare problem. Notably, the AF prevalence is expected to increase threefold in the next decade, commensurate with the aging population and the obesity epidemic. It is well known that obesity, defined by a high body mass index (BMI), plays an important role in the development of AF [1]. In multiple studies, obesity management has been shown to reverse the natural progression of AF. Specifically, the results of bariatric surgery studies indicated that a significant correlation could be found between AF reversal and weight loss [2,3]: animal models revealed that weight loss inversely changes cardiac structure and electrophysiology in obese sheep, resulting in reduced inflammation and fibrosis as well as an increased atrial effective refractory period and conduction velocity [4]. CA is an established AF treatment used to isolate the electrical signals arising from the pulmonary veins and thus restore sinus rhythm [5]. Interestingly, recent clinical evidence confirmed that AF patients with obesity undergoing CA are more likely to experience AF recurrence [3,4]: according to a meta-analysis by Akhtar et al., at follow-up more than 12 months after CA, there was a statistically significant decrease in AF recurrence in the case of weight loss (both >10% and <10% weight loss) compared to no weight loss [6]. Obesity is associated with cardiometabolic diseases through complex interactions between genes, epigenetics, environment, diet, and lifestyle. 

One primary etiological factor for the disruption of cardiac contractility and electrophysiological homeostasis might be miR dysregulation in obesity [7]. Adipose tissue is a major source of circulating miRs, which have recently been recognized as a new class of adipokines. A number of miRs derived from adipose tissue play an important role in adipocyte differentiation and obesity-associated inflammation and oxidative stress [7]. In addition to adipocytes, adipose tissue contains several types of immune cells capable of secreting hormones, cytokines, and other substances that regulate metabolic homeostasis [8]. Chronic myocardial inflammation acts as a key driver of cardiac fibrosis, which in turn leads to an accelerated loss of cardiac function and thus increased mortality [9,10]. While there are several factors leading to inflammation-associated myocardial fibrosis, obesity is one of the major disease drivers [11]. In particular, the epicardial adipose tissue (EAT), which is significantly enlarged in obese patients, has been recognized as a key mediator of cardiac inflammation and fibrosis contributing to heart failure and the development of AF [11,12]. In spite of the fact that weight management improves susceptibility to AF by addressing molecular mechanisms and factors that influence these processes, our understanding of these pathological processes remains limited.

We hypothesize that circulating miRs may be serve as predictors for AF recurrence after ablation. In this study, we aim to establish circulating miRs as biomarkers for guiding CA treatment strategies of AF in obesity, by linking circulating miRs to the key signaling pathways involved in pathological atrial structural and electrophysiological remodeling in this population.

## 2. Risk Factors for the Development of AF and the Possible Use of Circulating Associated miR as Biomarker for Pathological Remodeling and Therapeutic Reverse Modeling

It has been recognized that several established cardiovascular risk factors are independent predictors of the development of AF, including male sex, aging, obesity, excessive alcohol consumption, hypertension (HTN), and obstructive sleep apnea (OSA) [3,13].

Among those, obesity (body mass index ≥ 30 kg/m^2^), and HTN are the so-called modifiable risk factors and established described triggers for the growth of pro-inflammatory pathways, reactive oxygen species (ROS) culminating in pathways towards endothelial dysfunction (ED), atrial myocyte dilatation, and fibrosis, key pillars in the development of AF [14]. Thus, in addition to repeat CA, it is imperative to treat comorbid conditions that increase arrhythmia risk, i.e., obesity, HTN, and obstructive sleep-apnea (OSA) should be targeted and modified to increase the likelihood of treatment success. There is however, uncertainty regarding the exact mechanism through which weight loss reduces AF and recurrent AF post-CA (Figure 1).

### 2.1. Obesity, Adipose Tissue, and AF

Obesity, HTN, diabetes, and OSA are all associated with AF [1,13]. Over the past decade, obesity and AF, which are inextricably linked, have become increasingly prevalent throughout the world [1], leading to a higher prevalence of obesity among patients with AF than among healthy individuals. In addition to epicardial fat accumulation, obesity can cause systemic inflammation, and oxidative stress, all of which contribute to atrial enlargement, inflammatory activation, localized or diffuse myocardial fibrosis, and abnormal electrical conduction [15].

Weight loss has been shown to reduce the risk and reverse the natural progression of AF, which may be due to its anti-fibrotic and anti-inflammatory effects. In the clinical context, recent reports highlight that compared to normal weight, the increase in AF was significant (*p* < 0.01) for overweight, obese, and morbidly obese patients for newly-diagnosed AF, and for obese and morbidly obese patients for recurrent post-ablation AF [6,16].

However, the pathophysiological mechanisms associated with AF in obese patients are complex and remain unclear despite epidemiological studies establishing the role of obesity. Adipose tissue secretes a variety of pro-inflammatory and pro-fibrotic factors that can contribute to structural and functional remodeling of the left atrium and abnormal electrical conduction in the heart [1]. In addition to oxidative stress caused by adipose tissue, autonomic nerve activation in the cardiac autonomic ganglia (also known as ganglionated plexus) is also implicated in the occurrence of AF.

Importantly, it has been demonstrated in clinical studies that increased EAT, which is located between the visceral pericardium and epicardium [17], is associated with AF [18,19]: electrical remodeling, inflammation, fibrosis, and neurological factors contribute to the development of EAT-promoted AF. Serum MCP-1, IL-1, IL-6, soluble IL-6 receptor, and TNF levels in the blood are directly related to lower density and higher volume EAT [20]. Regional IL-1 levels in the EAT are risk factors for long-term AF [21]. The local conduction block and conduction delay caused by AF underpin the development of re-entrant arrhythmias [22]. YKL40 [23], cTGF [24], and other profibrotic cytokines/chemokines in the EAT are positively correlated with total collagen content in the left atrial myocardium, extensively reviewed in Shu et al. [1].

In addition to being pro-fibrotic and pro-inflammatory, EAT promotes oxidative stress [25] and Ca^2+^ homeostasis imbalance [26], both of which contribute to the formation and maintenance of AF.

Finally, the EAT is the location of the ganglionated plexus, which contains sympathetic and vagal fibers that regulate the autonomic nerves of the heart, and is closely associated with the onset and maintenance of AF [27].

Key miRs influencing adipogenesis and adipocyte differentiation by regulating lipid and glucose metabolism are miR-27a, miR-107, and miR-26a [16] (Figure 2).

### 2.2. Hypertension

Clinically, HTN and AF often coexist [28]. Uncontrolled or inadequately managed HTN damages target organs, leading to early mortality. These organs include the heart, brain, kidneys, and vasculature [29]. In terms of the heart, HF, ventricular and atrial arrhythmias, and LV hypertrophy are all linked to arterial HTN [3]. AF is significantly affected by HTN in terms of pathogenesis, management, and prognosis [30]. Epidemiological follow-up showed that more than 60% of patients with AF also suffer from HTN, and according to the Framingham study, HTN is associated with an increase in AF risk of 40–50% [31]. Considering the high incidence of AF in HTN, one could even argue that AF is another manifestation of HTN-related organ damage. Left atrial enlargement is associated with HTN and increased risk of recurrent AF after CA [32,33], while left atrial volume reduction following CA has been shown to predict sinus rhythm maintenance.

The American Heart Association report states that being overweight is a leading cause of HTN [34]. Obese participants had a 3.5-fold increased risk of high blood pressure, and fat buildup was a contributing factor in over 60% of high blood pressure cases [35]. In subjects of normal weight, the prevalence of HTN was 34% across all age groups; in subjects of obese weight, it ranged from 60 to 77% and increased in proportion to BMI [36].

These notions are further supported by experimental findings in animal models. Induced HTN causes interstitial fibrosis, myocyte hypertrophy and electromechanical remodeling [37], hallmarks of left atrial contractive dysfunction and AF.

In peripheral blood mononuclear cells, lower levels of miR-9 and miR-126 were linked to organ damage caused by HTN [38]: in the plasma of hypertensive cohorts, higher levels of miR-505, miR-21, and miR-122 were linked to HTN-associated endothelial dysfunction (Figure 2). On the other hand, HTN is one of the main risk factors for heart failure with the preserved ejection fraction (HFpEF), a common clinical syndrome with higher cardiovascular morbidity and mortality. For HTN-related HFpEF, miR-1233, -183-3p, -190a, -193b-3p, -193b-5p, -211-5p, -494, and -671-5p distinguished HT-associated HF from controls [38,39].

Renal disease caused by HTN is characterized by glomerular, tubule-interstitial, and vascular changes that can result in albuminuria or a decrease in glomerular filtration rate [38]. Two excellent independent sources of pathophysiological data pertaining to renal damage are miRs from urine and plasma extracellular vesicles. The first study of the urinary exosomal miRNome was developed in 2013 by Gildea et al. [40], who discovered 45 miRs linked to the blood pressure response to sodium. On the other hand, the highest levels of albuminuria were linked to lower urine exosomal miR-146a levels, according to a 2018 study conducted on patients with HTN who either had or did not have early renal damage [41].

To focus only on what matters most, more and more evidence indicates that miR-21 plays a prominent role in HTN [42], targeting many genes involved in the pathophysiology of HTN [42] like, e.g., TGFß receptors and signaling pathways and mitochondrial and endothelial regulatory pathways. Further miRs, including miR-9, 26, and 155, were mentioned to play important roles in target organ damage during HTN [43].

### 2.3. Sex Differences in AF

The clinical manifestation of AF varies between sexes. Men are more likely to experience AF, and it also tends to manifest earlier in life: the male sex is associated with a 1.5–2 times greater risk for AF than the female sex [44,45,46]. Across all age groups, men are more likely than women to have AF [46]. On the other hand, women typically experience more severe AF, which is frequently linked to a history of stroke and atrial fibrosis and excessive deposition of extracellular matrix proteins [47,48,49]. Thus, it would seem that men are not only more likely than women to develop AF but that they also need less adverse remodeling, which strongly suggests that there are differences in AF pathogenesis between the sexes. Despite the variety of mechanisms causing AF, all of them depend on the interaction of maintenance AF substrates with initiating triggers [50,51,52]. Specifically, only when the atria’s structural and electrophysiological characteristics are favorable can AF be sustained. Electrophysiological remodeling, which involves modifications to ionic currents such as K^+^, Na^+^, and Ca^2+^ currents, can cause this, leading to an altered action potential duration (APD) and effective refractory period (ERP) [47,51,53]. Importantly, connexins (Cx) play a crucial role in electrical conduction as well, because they enable the electrochemical coupling of nearby cells at the intercalated disks, which promotes cell-to-cell conduction. Cx40 and Cx43 are both expressed in the atria [54,55,56]. Patients with AF reportedly frequently show remodeling of the connexins. Notably, the conduction velocity can be slowed by their increased lateralization and decreased expression [54,55,56,57]. Furthermore, according to recent reports using mouse models, lateralization of connexins may enhance conduction heterogeneity by facilitating non-linear electrical impulse propagation through the atria [58,59,60]. The authors discuss that the Cx40 and Cx43 lateralization may encourage non-linear conduction and by this increase the male susceptibility to AF. To sustain AF, connexin remodeling in turn provides the required substrate [52,54,58]. In addition to irregularities in conduction, enlargement of the atrium and atrial fibrosis are indicative of structural remodeling in AF. In addition to slowing the conduction velocity through the atria, this structural remodeling is crucial for the upkeep of AF [61]. As a sex-related key difference, Thibault et al. report structural differences potentially fueling AF development: because their myocytes are larger than those of female animals, males were hypothesized to have a higher atrial mass than females, which provides a substrate for AF.

While the above observations argue strongly for an association of male sex and involvement of male sex hormones with AF, it is unclear whether female sex hormones could play a role in AF based on clinical data. Menopause is associated with an increase in the incidence of AF in females [62]. Despite a significant decrease in estrogens, changes in certain risk factors, such as body mass index (BMI), blood pressure, and cholesterol levels, may also be related to the higher incidence of AF during this time [63,64]. The available data suggest that female sex hormones do not appear to contribute to sex differences in AF occurring before menopause, at older age, or in AF that occurs regardless of pathological conditions and other known risk factors [65,66,67,68]. However, the involvement of estrogen may not be ruled out.

As detailed above, the major atrial gap junctional proteins Cx40 and Cx43 undergo numerous changes in expression and localization during AF, promoting sustained reentry, most prominently a pronounced lateralization preference of male AF-susceptible individuals. The miRs associated with sex differences in AF/gap junction-remodeling are miR-1 [69] and miR 208a-3p [70]. According to the Rotterdam study [71], general male-gender-associated miRs in AF-susceptible individuals related to atrial dilatation and stretch were determined to be miR-22 and -4798-3p (Figure 2).

### 2.4. Obstructive Sleep Apnea (OSA)

The presence of obstructive sleep apnea (OSA) increases the incidence, prevalence, and recurrence of AF. Furthermore, the treatment of AF, either using cardioversion or ablation, may be less effective in patients with untreated OSA [72,73,74].

OSA patients are predisposed to the formation of a substrate that raises the risk of AF. In a comprehensive study, the reduced expression of miR-21 and miR-208, miRs related to OSA, increased interstitial fibrosis, and inflammation have been described [75] (Figure 2).

### 2.5. Aging/Inflammaging

As with many other vital organs, the heart is susceptible to the effects of aging. Aging is associated with inflammation, as well as other coexisting comorbidities that are linked to age-related decline, such as cardiovascular disease, heart failure, and AF but also metabolic disease like type 2 diabetes [76,77,78,79]. Likewise, for the development of HTN, aging also plays a role. The lifetime risk of developing HTN is 90 percent even if you do not develop it by age 55 to 65 [80]. It is also known that inflammation contributes to AF. “Inflammaging”, a portmanteau of inflammation and aging, is defined as chronic, low-grade inflammation that accumulates over time and worsens with age, presumably contributing to the pathogenesis of a variety of age-related conditions, including AF [77,81,82,83,84]. There is evidence that inflammaging is caused by the senescence of the immune system’s cells, so-called immunosenescence [74].

Several reports have demonstrated a significant correlation between the systemic markers of inflammation, such as IL-1b, IL-6, IL-8, IL-18, C-reactive protein (CRP), interferon (IFN)-α and IFN-β, transforming growth factor β (TGF-β), and tumor necrosis factor (TNF), with the role of inflammation in AF in both human and experimental animal models. These significant cytokines are linked to inflammation and atherosclerosis, two conditions that worsen the health of the elderly and increase their risk of morbidity and death.

## 3. Converging Central Pathomechanisms Underlying AF Pathogenesis and Progression

### 3.1. Inflammation

There is increasing evidence that inflammation, both local and systemic, is a key factor in the development of AF as well as its progression towards persistent forms [85]. There is an essential role for tumor necrosis factor- (TNF-α), interleukin-2 (IL-2), and interleukin-1 (IL-1) in fibrosis. The transcription factor NFκB (nuclear factor kappa B) further exacerbates inflammation by regulating both inflammatory cytokines and ROS [47,86]. TNF-α may also support AF by regulating matrix metalloproteinase (MMP) activity and the degradation of ECM proteins [48]. NFκB activity, serum TNF- and IL-6 levels, and the collagen volume fraction in atrial tissue from AF patients with valvular heart disease are found to have a significant, positive correlation [49].

As pointed out above, as a result of immunosenescence, additional factors must be recognized in the elderly [87].

### 3.2. Endothelial Dysfunction (ED)

Being the first instance of vascular dysregulation, which in turn causes vessel atheromasia, a pro-thrombotic milieu, and cardiac arrhythmias, ED is exclusively linked to cardiovascular pathology [88].

A number of different cell phenotypes, including immune cells and fibroblasts, are disrupted by endothelium activation or dysfunction, which progresses toward anatomical milieu subversion and may play a major role in the development of AF and associated cerebrovascular accidents [89]. Consequently, systemic and local ED may be exacerbated by AF-induced hemodynamic and structural abnormalities [90,91]. Desantis [43] provides a thorough review of the various mechanisms that underlie AF pathogenesis and AF progression. These mechanisms include oxidative stress, pro-inflammatory signaling, genetic renin-angiotensin axis abnormalities, and intracellular Ca^2+^ overload. Reduced nitric oxide (NO) levels and increased ROS production are the results of the L-arg/eNOS synthase (endothelial nitric oxide synthase) pathway’s impaired activity during the early stages of endothelial activation/dysfunction. Through the creation of peroxynitrite, ROS further restrict the production of NO by encouraging vascular leukocyte adhesion and adding to the inflammatory environment in the vasculature [89], manifested by an upregulation of vascular cell adhesion molecule-1 (VCAM1) and intercellular adhesion molecule-1 (ICAM1) on the surface of endothelial cells [92,93,94].

miRs have been demonstrated to be intrinsically linked to ED and inflammation in AF [95]. miR-Let-7g preserves physiological endothelial functions by reducing monocyte adhesion, inflammation, and senescence while increasing angiogenesis by targeting genes in the TGF- and Sirt-1 signaling pathways. Endothelial activation and subsequent vascular lesions are caused by Let-7g function loss. miR-21 is involved in the modulation of the endothelial response to hemodynamic stress, in addition to its role in atrial remodeling. The master regulator of endothelial homeostasis and vascular integrity is miR-126, which targets VCAM-1 and proinflammatory mediators. By targeting tissue factors in monocytes, miR-126 has indirect antithrombotic properties and reduces vascular inflammatory responses (expression of VCAM-1 and fibrinogen, leukocyte counts) [96]. miR-155 is a direct regulator of eNOS expression and one of the most extensively studied inflammation-associated miRs. It reduces endothelial-dependent vasorelaxation by modulating TNF-induced eNOS suppression. In cardiomyocytes, activation of the NLRP-3 inflammasome promotes the progression of AF. The activation of the nucleotide-binding domain-like receptor protein 3 (NLRP3) inflammasome is suppressed by miR-135 [97]. Other miRs, e.g., miR-223, miR-197, and miR-10a, have been linked to vascular endothelial dysfunction in AF [43,98] (Figure 2).

### 3.3. AF and Mechano-Electric Coupling

According to De Jong et al. (2011) [99], chronic stretch results in atrial dilatation as well as diverse alterations in atrial architecture, including myocyte hypertrophy and fibrosis.

The atrial stretch-induced electrophysiological aftereffects comprise impaired conduction, alterations to the length and distribution of the atrial refractive period, and heightened susceptibility to AF [100,101]. Studies in both the clinic and the laboratory have revealed that AF is linked to the dysregulation of a number of miRs, which may serve as intermediaries in the structural and electrophysiological remodeling that keeps AF going [102]. Stretch and miR deregulation, brought on by underlying illnesses or risk factors, may encourage the atria’s structural and electrophysiological remodeling, which may aid in the onset and maintenance of AF. This in turn may perpetuate reentrant propagation by suitable substrates and give rise to ectopic activity by favoring early and delaying after depolarizations.

miR-1, miR-21, and miR-133 were the most frequently reported miRs to be dysregulated in human AF among those expressed in atrial tissue; these were followed by miR-25, miR-29/-29a, miR-208, and miR-590 [103]. For miR-21 and miR-208b, consistent or at least widespread upregulation patterns were seen, whereas miR-133 showed consistent downregulation patterns. Some miRs behaved in a way that was more unpredictable. In another report, there was a significant upregulation of miR-146 expression described, which regulates L-type Ca^2+^ channels in atrial cardiomyocytes [104]. Among the circulating miRs, the downregulated patterns of miR-21, miR-150, and miR-328 were more frequently reported as being deregulated in human AF [103] (Figure 2).

miR-21 contributes to the downregulation of Smad7, which strengthens the TGF-1/Smad signaling pathway and encourages the development of AF in both aortic arch patients and experimental study models [43].

Increased miR-328-3p has been linked to a higher risk of AF: according to the report, miR-328-3p acts as a mechanism underlying the atrial arrhythmogenic potential by targeting the genes encoding L-type Ca^2+^ channel proteins to reduce I_CaL_ density and increase AF vulnerability by shortening the duration of the atrial action potential [105,106].

On the other hand, additionally, miR133a prevents cardiac fibrosis by preventing TGF-β and other fibrosis-promoting factors from being expressed. As a result, it is an AF biomarker and a possible therapeutic target [29] that may help to avoid cardiac fibrosis and its side effects [107]. The increased expression of miR-26, one of the miRs implicated in the electrical mechanisms of AF, has been linked to dysregulation in pro-fibrillatory inward-rectifier potassium current changes.

Four other dysregulated miRs in patients with AF (miR-430-3p, miR-146b-5p, miR-630, and miR-367) have been suggested as potential viable biomarkers for the diagnosis of AF progression [108]. These miRs may impact the mTOR and Hippo signaling pathways.

There has been insight from experimental animal models: In addition, in the AF murine model, dysregulation in pro-fibrillatory inward-rectifier potassium current alterations has been linked to the elevated expression of miR-26 [89].

The dysregulation in pro-fibrillatory inward-rectifier potassium current changes in an AF murine model has been linked to the increased expression of miR-26, one of the miRs involved in the electrical mechanisms of AF [109].

### 3.4. Fibrosis in Obese AF Patients

An intricate relation of fibrosis and AF onset, as well as AF perpetuation, has been known for many years [110]. Fibroblasts constitute the main cellular mediators of extracellular matrix (ECM) production and degradation. Under physiological conditions, fibroblasts safeguard ECM homeostasis. Under pathological conditions however, fibroblasts are activated and undergo a phenotypic transition towards their secretory highly active myofibroblast phenotype, leading to a shift towards immoderate ECM deposition [10,111]. The interposition of insulating islets of ECM proteins within the normal histological architecture of the atria allows them to act as electrical conduction obstacles giving rise to arrhythmia. Furthermore, direct myofibroblast–cardiomyocyte coupling might further exert pro-arrhythmic effects [112,113]. There is a plethora of fibroblast-activating stimuli such as mechanical and oxidative stress, epigenetic modifications, and chronic inflammation [10,111]. The latter is particularly found in obese patients due to increased amounts of pro-inflammatory adipose tissue [111]. Regarding the heart, obesity is especially associated with increased deposition of EAT, which has been shown to have a direct influence on the adjacent atrial myocardium [11,12]. The proximity of the EAT to the atrial myocardium and the absence of a separating fascia allow for the diffusion of pro-inflammatory and profibrotic mediators from the EAT into the myocardium, promoting fibroblast activation, fibrosis, and AF [11]. There is evidence that weight loss, achieved by exercise, diet, bariatric surgery, or pharmacological intervention, leads to a reduction in EAT volume [114]. Therefore, weight loss and concomitant reduction in EAT-driven myocardial inflammation might be a promising therapeutic option to reduce atrial fibrosis and thus prevent AF.

The most relevant miR for the pathophysiology of AF processes of structural remodeling and TGF-regulated collagen synthesis are miR-21, miR-133, miR-30, and miR-29 (Figure 2).

### 3.5. The Relationship between Obesity, Oxidative Stress, and AF

The accumulation of oxidative stress in adipose tissue is one of the early events associated with obesity [1,109]. In obese animals and humans, oxidative stress is significantly increased in white adipose tissue compared to non-obese individuals [115]. There is a strong correlation between oxidative stress and the development of AF. An increase of 10% in redox glutathione is associated with a 30% increase in the risk of developing AF [116]. In animal models of atrial tachycardia and new-onset AF following cardiac surgery, antioxidants could prevent atrial electrical remodeling [1].

An increase in ROS, such as superoxide (O^2−^) and hydrogen peroxide (H_2_O_2_), has been linked to AF, which is often accompanied by a decrease in NO availability. As long-term effects, it has been found that Ang II, TGF-, and atrial stretch-mediated activation of the main enzymatic sources of ROS production, e.g., NAPDH oxidase 2 (NOX2) and NOX4, results in sustained oxidative stress, which stimulates myocyte apoptosis, atrial inflammation, fibrosis, and structural and electrical remodeling, which contributes to the development and persistence of AF [117].

In addition to atrial structural remodeling, electrical conductance is affected by ROS: ROS generated by oxidative stress trigger activity by enhancing late Na^+^ currents and provoking early depolarization [118]. Moreover, mitochondrial DNA damage caused by oxidative stress can cause Ca^2+^ overload in AF by modulating Ca^2+^-handling proteins or channels, which can lead to atrial electrical remodeling.

According to a literature review, miRs associated with AF might be regulated by ROS, suggesting that cellular stressors, such as ROS, may contribute to the development of AF [119]: ROS targets some key electrical remodeling-related molecules, such as CaMKII, RyR, LTCC, SERCA, and NCX, as well as an array of miRs with the potential to affect the arrhythmogenic process (e.g., targeting ion channels [108]): miR-Let-7e, -499, -1, -130a, -19, -145, -133, -328, -30, -23, and miR-21 to name the most prominent. There are four ROS-associated miRs associated with remodeling by modulating TGFβ-regulated collagen synthesis: miR-21, -133, -30, and miR-29 (Figure 2).

Thromboembolic complications in AF may also result from endocardial NO deficiency, caused by ROS production and eNOS uncoupling.

Furthermore, ROS stimulate the proliferation of atrial fibroblasts and promote the expression of inflammatory and pro-fibrotic factors, such as MMP9, p38, and c-Jun [117,120].

## 4. Synergistic Effects of Weight Loss and CA on miR Modulation in AF

Controlling body weight, and losing weight in particular, greatly reduces the risk of AF onset and maintenance and its associated morbidity and death [121]. In the LEGACY study (long-term effect of goal-directed weight management on atrial fibrillation cohort: a 5-year follow-up study), sustained weight loss led to a significant reduction in AF burden and was associated with beneficial structural remodeling [122]. These observational data indicate an important reversible component to the pathophysiology of the BMI/AF association and suggest a crucial role for weight reduction both before and after catheter ablation of AF to optimize the chances of success. Data from this group are in alignment with our analysis of the additive effect of weight loss and CA on arrhythmia recurrence [123].

Data generated by another group also support a synergistic effect of weight reduction both before and after CA of AF to optimize the chances of treatment success [6]. 

Particularly, it has been shown that weight loss reverses the miR dysregulation caused by obesity and inhibits adipogenesis, thereby further promoting weight loss [1,6,121]. Insight from animal models of obesity proved that the identified miR mitigated important metabolic changes, such as insulin resistance and lipid abnormalities, by modulating downstream miRs. Additionally, a few of them were even successful in lowering proinflammatory cytokines in adipose tissue. The availability of reliable matching preclinical data are however still scarce. Despite this lack of clinical evidence, the limited preclinical data helped to identify new potential biomarkers of metabolic alterations in obesity and related AF risk [121]:

Most prominently, the consistent downregulation of miR-26 was observed following dietary weight loss interventions in AF patients (Figure 3).

Among the molecular functions attributed to reduced expression of this miR in AF are (i) the increased expression of pro-fibrotic markers, (ii) a reduction in the inward rectifying K^+^ channel KNJ2 and dysregulation of L-type Ca^2+^ channels, contributing to arrhythmogenesis, and (iii) a reduction in the SELP protein, which increases the risk of thrombosis and an increase in the risk of cardioembolic strokes [124]. As a result of the formation of a thrombus in the left atrium, AF can increase the risk of ischemic stroke [125]. In patients with AF, the rate of ischemic stroke is approximately six times that of those without AF and varies greatly depending on the presence of co-existing cardiovascular disease. Apart from aging, obesity is one of the most important predisposing factors for AF and ischemic stroke [126]. However, the relationship between obesity, AF, and ischemic stroke remains unclear. There is current evidence that left atrial enlargement, which is strongly influenced by body weight, is an important precursor of AF. It increases stroke risk associated to AF through its effects on myocardial structure, autonomic function, ED, and inflammatory and pro-thrombogenic states [126].

Following a weight-loss dietary strategy, from all miRs investigated, miR-26 had the largest positively correlated fold-change (2.42, *p* = 0.05) between baseline and follow-up measurements [124]. Weight loss induced an upregulation of miR-26 expression, leading to consistent reverse remodeling of the above listed pathologic alterations [127,128].

The development of AF is closely associated with inflammation. It has been demonstrated that activation of the NLRP3 inflammasome promotes the progression of AF in cardiomyocytes. Its role in adipogenesis has been shown in isolated adipocytes [129]. miR-22 levels were reduced as a result of weight management. An increased level of circulating miR-22-5p and a defective regulation of intracellular calcium and cell-to-cell communication [130] are associated with AF in heart failure. A further study demonstrated that its overexpression promotes the obesity-induced miR-22/NLRP3 inflammasome axis [131]. Other than that, reduced miR-22, which is also critically involved in fibrotic disease by FGFR1 and FGFR2 induction, had consistently attenuated effects on the induction of both types of FGFR [132] following weight-loss interventions (Figure 3).

Supplementary miR-135 protects against AF recurrence following weight loss, by suppressing calcium-mediated NLRP3 inflammasome activation [97].

Further, in obesity-associated diseases including AF, an upregulation of miR-194, a regulator of mitochondrial dysfunction and cardiac injury, targeting GLP-1, is overexpressed [133].

From the miRs related to obesity-induced ED in AF, most prominently miR-21 appeared to be influenced by weight loss (Figure 3). miR-21 has been shown to affect eNOS through indirect signaling pathways such as AKT [134].

Decreased miR-126 levels have also been associated with AF and coexisting metabolic syndrome [135]. Weight loss increased the expression of miR-126, which reversed anatomical abnormalities such as hypertrophy and cardiac enlargement.

Reduced expression levels of miR-146 have been linked to obesity-associated increased severity of atherosclerosis in the lower limbs, the heart, and the carotid arteries, as well as AF. In particular, it has been demonstrated that miR-146b-p5p suppresses metalloproteinase inhibitor 4 (TIMP4), which in turn promotes atrial fibrosis in AF. Weight management was reported to support pro-atherogenic and pro-fibrotic reversal [104,136].

Data from experimental animal models show the following: In male C57BL/6 mice, cardiomyocyte apoptosis in AF was associated with the upregulation of miR-122 and decreased cardiomyocyte viability. Following miR-122 inhibition, the expression of the pro-apoptotic caspase-3 was significantly downregulated [137]. Further pathophysiological pathways attributed to miR-122 overexpression are fibrosis-related.

Additionally, miR-27b promotes atrial arrhythmia in a high-fat diet mouse model (HFD) [138]. The authors report that in HFD-stimulated atrial cardiomyocytes, miR-27b upregulation and concomitant Cx40 downregulation were observed. A subsequent luciferase assay demonstrated direct interaction between miR-27b and Cx40 3′. Interestingly, the inhibition of miR-27b by antisense oligonucleotides reversed the alterations caused by HFD stimulation [138]: for example, obesity was reversed, inflammatory cytokine expression (TNF-α, IFN-γ, IL-6, MCP-1, IL-1β, IL-17) was brought back to baseline, and insulin resistance was decreased in mice that lost weight (Figure 3).

The suitability of these miRs detected in animal studies as AF risk biomarkers in humans warrants further studies.

### Effects of CA on miR Modulation

miRs have been successfully employed as biomarkers for recurrence prediction after CA: miR-26 appears to be reduced in AF. Reportedly, in the treatment of AF, radiofrequency ablation reduced the expression of SELP protein by upregulating miR-26a/b. This reduced the thrombosis and cardioembolic stroke risk significantly [139] (Figure 3).

miR-21, a widely studied miR in association with AF, is associated with the presence of low-voltage areas in the left atrium. After ablation, the severity and recurrence of AF are correlated with the expression of miR-21 [140,141] (Figure 3).

The main endothelial miR, miR-22, and miR-126 exhibited long-lasting positive effects from CA (Figure 3):

Circulating miR-22 was found to be elevated in the serum of patients with heart-failure-related AF in a study examining the pattern of changes in multiple parameters for endothelial function, indicating that miR-22 is related to cardiovascular function (Figure 3) [141].

Numerous studies have demonstrated a positive correlation between thrombomodulin and hs-CRP levels and the recurrence of AF following cardioversion or ablation: Serum levels of high-sensitivity C-reactive protein (hs-CRP), a measure of vascular inflammation, and soluble thrombomodulin, which controls thrombosis on the surface of endothelial cells, had a strong correlation with primary (pri)-miR-126, the levels of which were restored after CA [142].

In cultured atrial cardiomyocytes, there was a marked upregulation of miR-146 expression, which regulates L-type Ca^2+^ channels, when compared to the control group in a study examining changes in the expression profile of circulating miR and the regulatory effect of AF-related miRs on ion channels [104].

In another report concentrating on novel biomarkers to improve risk management for patients with AF, miR-140 and miR-143 were identified [143]. According to the authors, miR-140 and miR-143 may serve as biomarkers, providing mechanistic and quantitative information on pathophysiology in everyday clinical practice with AF, such as possible profibrotic conditions in the left atrium, increasing stroke risk and reducing CA preference (Figure 3).

## 5. Concluding Remarks

Based on the data, the following conclusions can be drawn: following identification and functional analysis of miRs regulated by CA or weight loss in AF, a meaningful intersection as presented here was identified. The identified miRs could provide information about the pathway by which risk factors such as obesity and HTN contribute to the onset and maintenance of AF, and how CA can modulate this effect. Further research should be conducted on the identification of their targets and the possible therapeutic application of pri- and antagomirs.

### Limitations

First, a systematic review with corresponding registration as a study was not conducted in this study.

Second, combined data extraction from scientific review articles and clinical observations in compliance with various national standards has been prepared.

Third, it is important to interpret the results cautiously because there could be unanticipated selection bias in these analyses, as clinicians selected which patients received which form of CA or management therapy, respectively. While important associations were identified between miR, catheter ablation, and AF, any conclusions regarding causality should be avoided. It remains unclear whether targeting these miRs would result in a therapeutic effect beyond what is achieved by weight loss or the treatment of HTN.

Finally, it is critical to acknowledge the variety of miRs that have been identified throughout the included studies.

## 6. Materials and Methods

Relevant published articles were searched on PubMed databases from 2001 to present by using the following search criteria to retrieve articles and abstracts: (microRNA* or miR*) AND (Atrial fibrillation); depending on the chapter discussed in the review, we complemented this with AND risk (e.g., risk 1 = “obesity” OR risk 2 = “hypertension” OR risk 3 = “OSA”).

We focused on recent publications or findings validated by several independent studies. Studies were considered eligible on the basis of the following inclusion criteria: (i) miRNAs were evaluated in cardiomyocytes or endothelial cells or samples (blood) derived from patients with AF, including publicly available breast cancer cohorts; or samples (blood or tissue) derived from animal models of AF; (ii) the relationship between the studied miRNAs and AF was investigated; or (iii) the relationship between miRNAs and outcome or therapy response in AF was examined.

Articles were excluded if they met one of the following criteria: (i) published articles were retracted articles or comments; (ii) lack of key information on AF biology, prevention, or prediction of therapy response; and (iii) manuscripts reported on cardiovascular disease types other than AF.

## Figures and Tables

**Figure 1 ijms-25-04689-f001:**
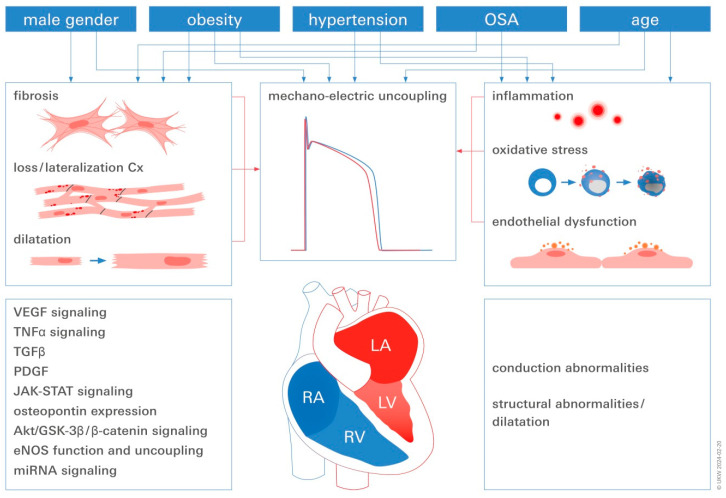
Major risk factor-related mechanisms in AF: This figure shows the relationship between atrial remodeling and AF and the mechanistic contributors that have been shown to be caused by different risk factors. It should be noted that risk factors are not independent; rather, they frequently contribute to the development of other risk factors. For example, obesity increases the risk of hypertension, diabetes mellitus, and OSA, and both conditions raise the risk of hypertension. RA stands for right atrium, RV for right ventricle, OSA for obstructive sleep apnea, Cx for connexin, LA for left atrium, and LV for left ventricle.

**Figure 2 ijms-25-04689-f002:**
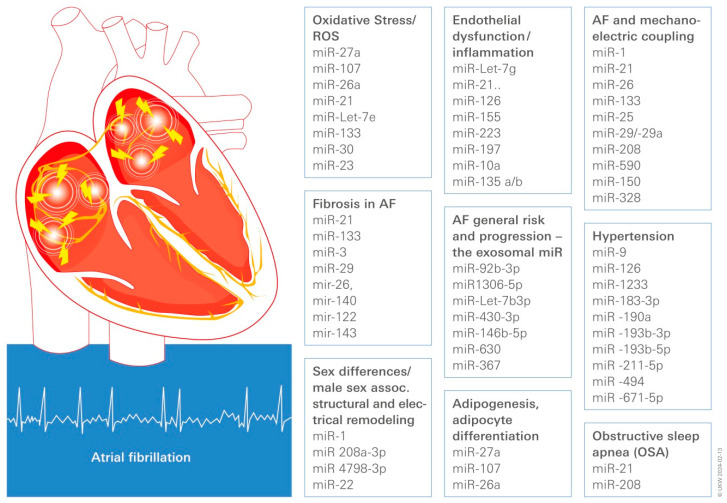
Diagrammatic representation illustrating the miRs most commonly linked to HTN, OSA, ED, inflammation, fibrosis, adipogenesis, oxidative stress, male gender, and mechanoelectric uncoupling, which may be involved in the development and progression of AF.

**Figure 3 ijms-25-04689-f003:**
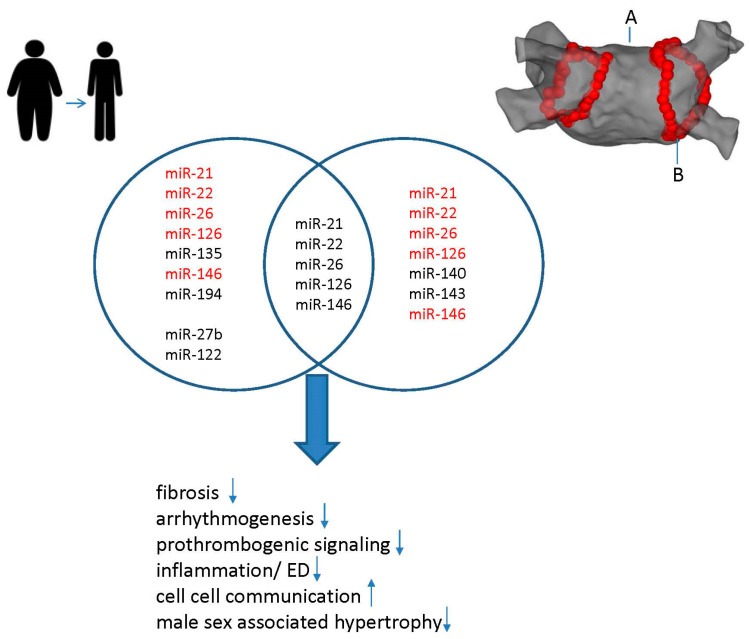
Evidence of synergistic effects of weight loss and CA for AF management. Proposed effects of targeting miR identified at the intersection of weight loss (**left**) and CA (**right**) in AF (upregulation = arrow up; downregulation = arrow down): Following weight loss and CA, the heart’s structure and electrophysiology undergo reverse remodeling, which is characterized by an increase in conduction velocity, an increase in intercellular communication, and a decrease in fibrosis and coagulation as well as inflammation. (A) Left atrium. (B) Ablation lesions around pulmonary veins.

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
