# Peer review of "Synergistic Effects of Weight Loss and Catheter Ablation: Can microRNAs Serve as Predictive Biomarkers for the Prevention of Atrial Fibrillation Recurrence?"

_ijms, 2024, doi:10.3390/ijms25094689_

Round 1

Reviewer 1 Report

Comments and Suggestions for Authors

The present review is extremely detailed and contains too many information and data on risk factors for AF, independent from obesity and weight loss.

The authors refer in the first 3 parts in information regarding risk factors for AF development and the role of miRs as biomarkers in a difficult to follow way (including sex differences and inflammaging).

In the 3rd part they refer to pathogenic mechanisms related to AF, again in a difficult to follow way, focusing on essentially irrelevant and non specific to weight loss parameters.

In general, the concept is not clear cut and straight forward. Report to miRs, weigh loss and ablation is practically present only in the 4th part of the manuscript and the first 3 parts are focused in general in AF pathogenesis.

I am not sure I can review back this paper.

Author Response

Response reviewer 1

We appreciate the reviewer's feedback and have taken steps to enhance the clarity and focus of the manuscript. Here are our responses addressing the specific concerns:

Detail and Scope: We recognize that the review contains extensive information on risk factors for AF beyond obesity and weight loss.

We would like to justify this as follows: the purpose of this review is to present and extrapolate/report experimental and clinical data that demonstrate the role of miRNA in modulating electrical and structural remodeling of the atrium in atrial fibrillation (AF and AF ablation) as a function of obesity and weight loss. Our strategy: This review presents and extrapolates experimental and clinical data that demonstrate miRNA modulation of electrical and structural remodeling of the atrium in atrial fibrillation (AF and AF ablation) in the context of obesity and weight loss. The objective of this study is to identify novel or pharmacologically druggable major signaling pathways and miRNAs that decrease AF recurrence, resulting from weight loss in obesity, as demonstrated in clinical observations by Akthar et al. [DOI: 10.1111/jce.16090 ]. The objective of this study is to identify new or pharmacologically targetable signaling pathways that decrease recurrence of AF in obese individuals, as demonstrated by Akthar et al. in their clinical observations. We believe that the comprehensive and detailed description of individual miRNAs in this review might provide a comprehensive guideline to motivate future, more standardized clinical studies focusing on the miRNAs listed in our review with the aim of preventing the recurrence or manifestation of AF.

Difficulty in Follow-up: We apologize for any confusion caused by the previous organization of the manuscript. In response, to facilitate reading, we revised the organization of the sections and paragraphs to provide more clarity and improve the flow of reading. We streamlined the content while maintaining comprehensive coverage of the topic. Additionally, we included a new section elaborating how the study design was planned and how were the articles selected for consistency and coherence.

Relevance of Pathogenic Mechanisms: Since the described pathophysiological mechanisms of fibrosis, mechanoelectric uncoupling, oxidative stress, and inflammation, have all been linked to the metabolic syndrome 1 the alterations in miRNA patterns of these specific pathophysiological mechanisms were examined.

Understanding the importance of focusing on pathogenic mechanisms relevant to weight loss and catheter ablation in AF might be mandatory. For many pathologies to be resolved or even prevented, obesity and other modifiable risk factors should be addressed with great concern.We believe that the practice of preventive medicine should be encouraged in all branches of medicine to the maximum extent possible.This is especially true in cases involving cardiac conditions and their recurrence. Therefore, in addition, we now included information on how those pathophysiological mechanisms can be clinically correlated with parameters such as weight and BMI.

Clarity and Conceptual Focus: We reorganised the sections and paragraphs to improve the reading flow and provide greater clarity based on the reviewers' suggestions resulting in a more focused and conceptually clear presentation. Furthermore, we made sure that every acronym was introduced correctly.

  1. Shu H, Cheng J, Li N, Zhang Z, Nie J, Peng Y, Wang Y, Wang DW, Zhou N. Obesity and atrial fibrillation: A narrative review from arrhythmogenic mechanisms to clinical significance. Cardiovascular diabetology. 2023;22:192

Reviewer 2 Report

Comments and Suggestions for Authors

The authors have described a very important topic, which can change the current clinical scenario of the atrial fibrillation patients. Obesity and other modifiable risk factors for any disease should be of major concern to resolve many pathologies or even to prevent from occurring. Preventive medicine should be encouraged as much as possible across all branches of medicine. And especially in cardiac cases. The diagrammatic representations are appreciated, it increases the interest and gives a better understanding to the reader.

That being said, I still have a few suggestions for the authors. 

In the title you have specified the term recurrence, but during the entire text you have emphasized more on prevention. Kindly justify.

The way of organising could be improved. So that the information provided in each section can flow in connection.

Regarding the underlining part I don't see the importance of it, I would suggest to remove it. 

in the abstract and later in the entire you have to state the full form of any abbreviation used for the first time. Which has been missed at some places, kindly correct it. 

You can elaborate how the study design was planned and how were the articles selected for making a review. 

The article by Rosca CI et al can help put some more light to the importance of the topic (https://www.mdpi.com/2227-9059/11/7/2012)

Throughout the text remove hyperlinks and try to thoroughly revise the text, there are some typos and grammatical errors. Correct them. 

You have explained the underlying mechanisms of pathophysiology, but I would suggest you to add also how to clinically correlate them apart from weight and BMI parameters. Or state in the limitations of the study. 

Author Response

ijms-2911942R1 - Point by Point Response to the Reviewer Comments

Reviewer #2

The authors have described a very important topic, which can change the current clinical scenario of the atrial fibrillation patients. Obesity and other modifiable risk factors for any disease should be of major concern to resolve many pathologies or even to prevent from occurring. Preventive medicine should be encouraged as much as possible across all branches of medicine. And especially in cardiac cases. The diagrammatic representations are appreciated, it increases the interest and gives a better understanding to the reader.

Answer: We thank the reviewer for their time and effort spent on the evaluation of our manuscript and for their generally positive feedback.

Comment 1. In the title, you have specified the term recurrence, but during the entire text you have emphasized more on prevention. Kindly justify.

Answer to Comment 1. The reviewer raised a valid point. The main topic of our ms is in fact recurrence prevention, thus, we want to identify and establish microRNAs which might be suitable to serve as Predictive Biomarkers for Atrial Fibrillation Recurrence. We adapted the title of the study accordingly.

Comment 2. The way of organising could be improved. So that the information provided in each section can flow in connection. The article by Rosca CI et al can help put some more light to the importance of the topic (https://www.mdpi.com/2227-9059/11/7/2012)

Answer to Comment 2. According to the reviewers’ suggestion, we revised the organization of the sections and paragraphs to provide more clarity and improve the flow of reading.

Comment 3. Regarding the underlining part I don't see the importance of it, I would suggest to remove it. 

Answer to Comment 3. We removed all underlining parts we could identify.

Comment 4. In the abstract and later in the entire [manuscript] you have to state the full form of any abbreviation used for the first time. Which has been missed at some places, kindly correct it.

Answer to Comment 4. We thank the reviewer for this critical comment. All abbreviations are now correctly introduced.

Comment 5. You can elaborate how the study design was planned and how were the articles selected for making a review.

Answer: An additional section “materials and methods” was introduced to explain the study deseing and how the articles were selected.

Comment 6. Throughout the text, remove hyperlinks and try to thoroughly revise the text, there are some typos and grammatical errors. Correct them.

Answer to Comment 6. We apologize for the typos. We went through the whole manuscript and carefully corrected all typos and grammatical mistakes. Concerning the hyperlinks, these are the Endnote references, and thus cannot be removed.

Comment 7. You have explained the underlying mechanisms of pathophysiology, but I would suggest you to add also how to clinically correlate them apart from weight and BMI parameters. Or state in the limitations of the study. 

Answer to Comment 7:

While important associations were identified between miR, catheter ablation and AF, any conclusions regarding causality should be avoided. It remains unclear whether targeting these miR would result in a therapeutic effect beyond what is achieved by weight loss or treatment of HTN. We have modified the limitations section accordingly.
